# DIFFERENTIABLE GRAPH OPTIMIZATION FOR NEURAL ARCHITECTURE SEARCH

## ABSTRACT

In this paper, we propose Graph Optimized Neural Architecture Learning (GOAL), a novel gradient-based method for Neural Architecture Search (NAS), to find better architectures with fewer evaluated samples. Popular NAS methods usually employ black-box optimization based approaches like reinforcement learning, evolution algorithm or Bayesian optimization, which may be inefficient when having huge combinatorial NAS search spaces. In contrast, we aim to explicitly model the NAS search space as graphs, and then perform gradient-based optimization to learn graph structure with efficient exploitation. To this end, we learn a differentiable graph neural network as a surrogate model to rank candidate architectures, which enable us to obtain gradient w.r.t the input architectures. To cope with the difficulty in gradient-based optimization on the discrete graph structures, we propose to leverage proximal gradient descent to find potentially better architectures. Our empirical results show that GOAL outperforms mainstream black-box methods on existing NAS benchmarks in terms of search efficiency.

## 1 INTRODUCTION

Neural Architecture Search (NAS) methods achieve great success and outperform hand-crafted models in many deep learning applications, such as image recognition, object detection and natural language processing (Zoph et al., 2017; Liu et al., 2019; Ghiasi et al., 2019; Chen et al., 2020). Due to the expensive cost of training-evaluating a neural architecture, the key challenge of NAS is to explore possible good candidates effectively. To cope with this challenge, various methods have been proposed, such as reinforcement learning (RL), evolution algorithm (EA), Bayesian optimization (BO) and weight-sharing strategy (WS), to perform efficient search (Zoph & Le, 2016; Real et al., 2019; Hutter et al., 2011; Liu et al., 2019; Guo et al., 2019).

While the weight-sharing strategy improves overall efficiency by reusing trained weights to reduce the total training cost, zeroth-order algorithms like RL, EA and BO employ black-box optimization, with the goal of finding optimal solutions with fewer samples. However, the search space of NAS is exponentially growing with the increasing number of choices. As a result, such huge combinatorial search spaces lead to insufficient exploitation of black-box learning framework (Luo et al., 2018).

Another line of research has been focused on formulating the NAS search space as graph structures, typically directed acyclic graphs (DAGs), and then the search target is cast as choosing an optimal combination of the nodes and edges in the graph structure (Pham et al., 2018; Liu et al., 2019; Xie et al., 2019). However, existing methods tend to perform the optimization in the indirect manner using black-box optimization. In contrast, we aim to explicitly model the search space as graphs and optimize graph structures directly. We thus propose Graph Optimized Neural Architecture Learning (GOAL), a novel NAS approach combined with graph learning for efficient exploitation, as briefly shown in Fig.1. Unlike other black-box approaches, we use a differentiable surrogate model to directly optimize the graph structures. The surrogate model takes a graph structure corresponds to a neural architecture as input, and predicts a relative ranking score as the searching signal. We then apply gradient descent on the input graph structure to optimize the corresponding architecture, which attempts to obtain a better predicted ranking score. As we optimize the surrogate model and the architectures iteratively, the optimal architectures could be typically obtained after a few iterations. In particular, to cope with the difficulty of using gradient-based optimization on the discrete graph structure, we adapt the proximal algorithm for allowing us to optimize discrete variables in a

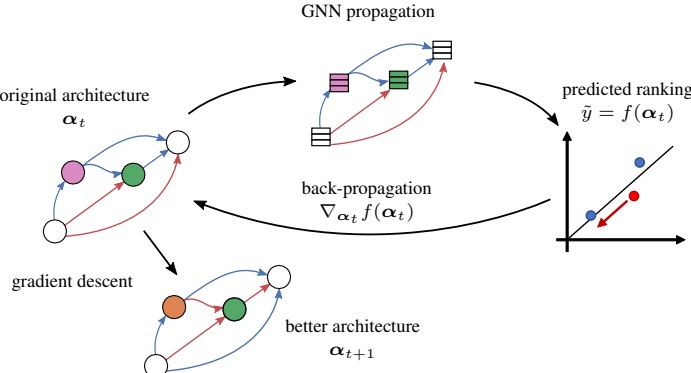

Figure 1: Overview of the GOAL steps. A GNN-based surrogate model $f$ predicts the ranking score $\tilde{y}$ of the original architecture $\boldsymbol{\alpha}_t$, then back-propagating $\tilde{y}$ through the GNN model to compute the gradient w.r.t. $\boldsymbol{\alpha}_t$. A better architecture $\boldsymbol{\alpha}_{t+1}$ is obtained by proximal gradient descent.

differentiable manner (Parikh et al., 2014; Bai et al., 2019; Yao et al., 2019). We build the surrogate model with Edge Conditional Convolution (ECC) (Simonovsky & Komodakis, 2017), a variant of Graph Convolutional Network (GCN) (Kipf & Welling, 2017), to handle the graph representation of various search spaces.

Our empirical results on existing NAS benchmarks with different search spaces demonstrate that, GOAL outperforms exist state-of-the-art black-box optimization baselines and neural surrogate model based methods by a large margin in terms of the search efficiency.

The main contributions of this paper are summarized as follows:

- We propose a differentiable surrogate model for ranking neural architectures based on GNN, which takes advantage of the graph structure of neural architectures and guides the architecture search efficiently.

- We present GOAL, a novel gradient-based NAS sample-efficient approach with the assistance of the proposed surrogate model. Comparing to exist algorithms with GNN surrogates, GOAL makes full use of the learned representation by jointly optimizing the GNN model and candidate architectures and performing efficient exploitation within graph-structured search spaces.

- Our empirical results demonstrate that the GOAL significantly outperforms existing state-of-the-art methods in various search spaces settings.

## 2 RELATED WORKS

### 2.1 NEURAL ARCHITECTURE SEARCH

From the earliest explorations on automatic neural network designing to recent NAS trends, the NAS problem developed from hyper-parameter optimization, becoming a more challenging task due to the inherent complexity of its search space (Bergstra et al., 2013; Elsken et al., 2018). Existing popular approaches include various kinds of algorithms: reinforcement learning (Zoph & Le, 2016; Zoph et al., 2017; Pham et al., 2018), evolution algorithm (Real et al., 2019), Bayesian optimization (Falkner et al., 2018; White et al., 2019), monte carlo tree search (Wang et al., 2019b;a), gradient based methods (Liu et al., 2019; Luo et al., 2018), etc. There are also some works employ surrogate models to predict the performance of architectures before training to reduce the cost of architecture evaluation (Liu et al., 2018; Wen et al., 2019; Wang et al., 2019b). Some most recently parallel works even tries to improve black-box optimization methods like Bayesian optimization by efficient surrogate predictors (White et al., 2019; Shi et al., 2019). Existing gradient-based methods usually employ weight-sharing based relaxation or use encoder-decoder to optimize a continuous hidden space (Liu et al., 2019; Luo et al., 2018). These approximations cause biased model criterion and generation, which can harm the final performance (Yu et al., 2019; Yang et al., 2020). In contrast, our method directly optimizes the discrete architectures, avoids the biased model criterion.

Due to the complex search settings and expensive evaluation cost, NAS works are hard to evaluate and reproduce fairly (Li & Talwalkar, 2019; Yang et al., 2020). *NAS-Bench-101* and *NAS-Bench-201* are proposed to provide fair and easy benchmarks for NAS algorithms (Dong & Yang, 2020; Ying et al., 2019).

Taking advantage of the development of graph neural networks (GNNs), several recent works show that the power of GNNs can also benefit the NAS task (Shi et al., 2019; Wen et al., 2019). However, these works only employ GNNs as a powerful black-box predictor. In contrast, we make full use of the learned representation of GNN by back-propagating through the GNN model and performing gradient-based optimization on the input graphs.

## 2.2 GRAPH STRUCTURE LEARNING

Graph neural network (GNN) is a kind of neural model for extracting features from graph-structured data (Zhou et al., 2018). The most popular family of GNNs is the Graph Convolutional Network (GCN), which propagates the features of nodes in the spectral or spatial domain (Kipf & Welling, 2017; Morris et al., 2019). Since the vanilla GCN can only handle monomorphic graphs with single type of edge, many variants have been proposed to handle more complex graphs (Schlichtkrull et al., 2018; Simonovsky & Komodakis, 2017).

As the normal GNN pipeline requires an available fine-grained graph structure, recent works present approaches to optimize the graph structure and GNN model jointly (Chen et al., 2019; Franceschi et al., 2019). The graph structures could be either constructed from scratch or fine-tuned from a sub-optimal graph with the GNN model. We follow this manner, iteratively optimize the graph of neural architectures, guided by the GNN model.

## 3 PRELIMINARY: NEURAL ARCHITECTURE SEARCH SPACES

While our method can be potentially applied more generally, in this paper we focus on the most commonly used *cell-based* search spaces of convolutional neural networks (CNNs) (Zoph et al., 2017; Pham et al., 2018; Liu et al., 2019; Ying et al., 2019; Dong & Yang, 2020). A cell is a basic building block of architectures represented as a DAG, which consists of several basic neural operations and their connectivity relationship as shown in Fig.2. Popular basic operations include different kinds of convolutions and poolings, such as $1 \times 1$ convolution, $3 \times 3$ convolution, $3 \times 3$ max pooling, etc. A complete CNN architecture is thus formed by stacking several repeats of the same cell.

There are different kinds of representations to form an architecture by a DAG. Fig.2 shows two typical representations. It is non-trivial to convert from one to each other, since the corresponding architectures formed by different representations do not coincide.

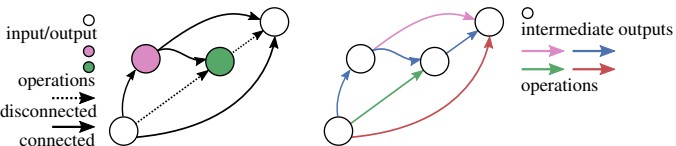

Figure 2: Two typical architecture representations as DAGs. Left: nodes as operations, edges as connections; right: nodes as intermediate outputs, edges as operations.

The cell-based search spaces are usually heuristically constrained by limiting the max number of nodes and edges, or the degree of each node, which corresponds to the number of operations and connections in a cell (Liu et al., 2019; Ying et al., 2019). This brings different feasible sets in our optimization in Sec.5.1.

## 4 LEARNING TO RANK ARCHITECTURES BY GRAPH

We first design the GNN model which guides our optimization on neural architectures. Denoting a neural architecture setting as $\boldsymbol{\alpha}$, we train a differentiable model $f(\boldsymbol{\alpha})$ which predicts a score $\tilde{y}$ to

indicate the real performance $y$, e.g. accuracy on the image classification task, of $\boldsymbol{\alpha}$. We build the model as GNN by Edge Conditional Convolution (ECC) layers (Simonovsky & Komodakis, 2017). While the GNN model itself is fully differentiable, it is possible to obtain gradient w.r.t $\boldsymbol{\alpha}$ from $f(\boldsymbol{\alpha})$ to optimize the architecture.

## 4.1 GRAPH REPRESENTATION OF NEURAL ARCHITECTURES

As described in Sec.3, mainstream NAS works search architectures in search spaces which could be represented as directed acyclic graphs (DAGs). We form these search spaces as a unified perspective.

Denoting $G = (V, E, \mathcal{T}, \mathcal{A})$ as the graph corresponding to an architecture $\boldsymbol{\alpha}$, where $V$ is the set of nodes, $E$ is the set of edges, $\mathcal{T}$ and $\mathcal{A}$ denote the types of nodes and edges respectively. For a predefined search space, we can form each architecture in the space by fixing $V$ and $E$ as a fully-connected DAG, where each node $i$ is connected to each node $j$ for any $i < j \leq |V|$, and assign proper $\mathcal{T}$ and $\mathcal{A}$. As the connectivity between nodes can also be denoted as an edge type (*i.e.* denoting as a *disconnected* type, as shown in the left of Fig.2), we can determine both operation types and connectivity of the graph by $\mathcal{T}$ and $\mathcal{A}$. The neural architecture setting $\boldsymbol{\alpha}$ is therefore formed by $\boldsymbol{\alpha} = (\mathcal{T}, \mathcal{A})$, consisting of the types of nodes and types of edges for graph structure. In particular, we form each component of $\mathcal{T}$ and $\mathcal{A}$ as a one-hot vector, which encodes a categorical selection.

To build the surrogate model dealing with various nodes and edges types in the graph representation of neural architectures, we employ the ECC, a variant of GCN, to extract the features $\boldsymbol{h}_i$ of each node $i$ in the graph (Simonovsky & Komodakis, 2017). For each node $i$ in the directed graph $G$, the $l$-th layer of a $L$-layers ECC propagates features $\mathbf{h}_j^{\ell-1}$ of the previous layer from node $j \in \mathcal{N}(i)$ to node $i$, where $\mathcal{N}(i)$ denotes the predecessors set of node $i$. The ECC learns a kernel generator $k_e : \mathbb{A} \mapsto \mathbb{R}^{d_\ell \times d_{\ell-1}}$, which maps the edge attribute space $\mathbb{A}$ to a real-valued $d_\ell \times d_{\ell-1}$ filter for the graph convolution, where $d_\ell$ indicates the number of channels in the $\ell$-th convolution layer. Thus the graph convolution is formed by:

$$\mathbf{h}_i^\ell = \boldsymbol{\Theta}^\ell \mathbf{h}_i^{\ell-1} + \sum_{j \in \mathcal{N}(i)} k_e^\ell(\mathbf{a}_{ji}) \mathbf{h}_j^{\ell-1} \tag{1}$$

where $\mathbf{a}_{ji}$ is the attribute, *i.e.* categorical selection as one-hot vector, of the edge from node $j$ to node $i$, $\boldsymbol{\Theta}$ is a learnable parameter.

For high-level prediction upon the whole graph, we compute the feature vector $\mathbf{h}_G$ of the whole graph $G$ by taking the weighted sum of each node features of the last layer (Li et al., 2016):

$$\mathbf{h}_G = \sum_i \frac{\exp(\mathbf{w}\mathbf{h}_i)}{\sum_j \exp(\mathbf{w}\mathbf{h}_j)} \mathbf{h}_i \tag{2}$$

where $\mathbf{w}$ is a learnable parameter to compute the weight for features of each node. Followed by fully-connected layers that take $\mathbf{h}_G$ as input, we can get the final prediction $\tilde{y}$ as an indicator score of the input architecture.

## 4.2 OBJECTIVE FUNCTION

Since we only care about the relative order to gain better architectures, we use the pair-wise rank loss for our training objective function (Burges et al., 2005). Let $D = \{(\boldsymbol{\alpha}_i, y_i)\}$ denotes the dataset which contains all evaluated architecture $\boldsymbol{\alpha}_i$ and corresponding performance $y_i$, $\tilde{y}_i = f(\boldsymbol{\alpha}_i)$ be the predicted score, the loss $\mathcal{L}$ is formed by:

$$\mathcal{L} = \sum_{\substack{(\boldsymbol{\alpha}_i, y_i),(\boldsymbol{\alpha}_j, y_j) \in D \\ i \neq j}} \log\left(1 + e^{-(\tilde{y}_i - \tilde{y}_j) \cdot \mathrm{sign}(y_i - y_j)}\right). \tag{3}$$

Thus the prediction $\tilde{y}$ could be used to select the best architectures.

## 5 SEARCH BY JOINT OPTIMIZATION

We thus present our method for searching optimal architectures by gradient. We jointly optimize the surrogate model and the candidate architectures alternately to find an optimal final result. For allowing gradient-based optimization on the discrete graph structures corresponding to neural architectures, we introduce the proximal gradient descent on the graph structures. Then we present the complete search algorithm.

### 5.1 PROXIMAL GRADIENT DESCENT ON ARCHITECTURES

Since the learned surrogate $f$ in Sec.4 is differentiable, we want to perform gradient descent on $\boldsymbol{\alpha}$ by the predicted $\tilde{y}$ to find better architecture. However, the edge type selection in the searching phase is discrete, bringing hardness to optimize $\boldsymbol{\alpha}$ directly by gradient descent. We thus introduce the proximal optimization to perform gradient-based optimization on a discrete feasible set (Parikh et al., 2014; Yao et al., 2020; Bai et al., 2019). Assume the learnt $f$ outputs lower $\tilde{y}$ for better architecture $\boldsymbol{\alpha}$, our search could be formed as a constrained optimization $\arg\min_{\boldsymbol{\alpha}} f(\boldsymbol{\alpha})$, s.t. $\boldsymbol{\alpha} \in \mathcal{S}$, where $\mathcal{S}$ is the feasible set depends on the search space. Typically, for the graph representation $G = (V, E, \mathcal{A}, \mathcal{R})$ corresponds to $\boldsymbol{\alpha} = (\mathcal{A}, \mathcal{R})$, $\mathcal{S}$ requires the $\mathcal{A}$ and $\mathcal{R}$ to be discrete categories for operation selection. Recall that we form each component in $\mathcal{A}$ and $\mathcal{R}$ as one-hot vectors, we update $\boldsymbol{\alpha}$ iteratively with a continuous approximation $\bar{\boldsymbol{\alpha}}$, whose components are real-valued continuous vectors, by following iterative steps:

$$
\begin{aligned}
\boldsymbol{\alpha}_t &= q(\bar{\boldsymbol{\alpha}}_t) \\
\bar{\boldsymbol{\alpha}}_{t+1} &= \bar{\boldsymbol{\alpha}}_t - \mu \nabla_{\boldsymbol{\alpha}_t} f(\boldsymbol{\alpha}_t)
\end{aligned}
\tag{4}
$$

where $\mu$ is the gradient step size; $q(\cdot)$ is the proximal operator maps continuous variables to the feasible set $\mathcal{S}$, which satisfies $q(\bar{\boldsymbol{\alpha}}) = \arg\min_{\boldsymbol{\alpha}} \|\boldsymbol{\alpha} - \bar{\boldsymbol{\alpha}}\|$, s.t. $\boldsymbol{\alpha} \in \mathcal{S}$. Eqn.(4) is known as a limiting case of lazy proximal step (Bai et al., 2019; Xiao, 2010). Practically we harden each continuous component vector in $\boldsymbol{\alpha}$ to the nearest one-hot categorical vector by setting the max-valued component to 1 and others to 0.

### 5.2 SEARCH PROCEDURE

---

**Algorithm 1:** GOAL: Graph Optimized Neural Architecture Learning

---

**Input:** Random initialized architecture candidates pool $P = \{\bar{\boldsymbol{\alpha}}_0, \bar{\boldsymbol{\alpha}}_1, ..., \bar{\boldsymbol{\alpha}}_N\}$; max time cost budget $T$; max number $K$ for new produced architectures in each step; random initialized surrogate model $f$.

Train and evaluate each $q(\bar{\boldsymbol{\alpha}})$ corresponds to $\bar{\boldsymbol{\alpha}} \in P$ ;

**while** *unsatisfied terminating condition* **do**

> Train $f$ with $P$ using loss function $\mathcal{L}$ in Eqn.(3);
>
> $\tilde{P}_{\text{new}} \leftarrow$ Obtain new $\bar{\boldsymbol{\alpha}}$ for each $\bar{\boldsymbol{\alpha}}_i \in P$ by Eqn.(4);
>
> $P_{\text{new}} \leftarrow \bar{\boldsymbol{\alpha}} \in \tilde{P}_{\text{new}}$ with the best $K$ predicted ranking score;
>
> **for** $\bar{\boldsymbol{\alpha}}_i \in P_{\text{new}}$ **do**
>
>> Train and evaluate the architecture corresponding to $q(\bar{\boldsymbol{\alpha}}_i)$;
>
> **end**
>
> $P \leftarrow P \cup P_{\text{new}}$;

**end**

**Result:** The best architecture in $P$.

---

The essential of the search is a joint optimization process on the architectures and the surrogate model $f$. A complete algorithm description is presented in Alg.1.

The search process starts at a small batch of random sampled architecture continuous approximations, which we denote as the architecture candidates pool $P = \{\bar{\boldsymbol{\alpha}}_i\}$. We train and evaluate each architecture $\boldsymbol{\alpha}_i$ corresponds to $q(\bar{\boldsymbol{\alpha}}_i)$, then train the initial surrogate model $f$ with the evaluated architectures as training data.

In each iteration of the search loop, we optimize each $\bar{\alpha}_i \in P$ by Eqn.(4) to obtain a potentially better architecture approximation. The new generated approximations with the best $K$ predicted ranking scores will be evaluated and collected to enlarge the candidates pool $P$. Then the enlarged pool $P$ is used to further improve the surrogate model $f$, which enable more precise prediction and gradient for architecture exploitation.

The process terminate at a pre-defined condition, *e.g.* the total training cost budget. We choose the best architecture in $P$ as the final result.

### 5.3 WEIGHT-SHARING STRATEGY

Weight-sharing is a powerful approach for reducing the total training time on architecture evaluation (Pham et al., 2018). GOAL can easily take benefits from weight-sharing to further improve the overall efficiency.

Our weight-sharing pipeline follows Li & Talwalkar (2019) and Guo et al. (2019) which first train a super-net model, then evaluate architectures as substructures in the super-net to search the best architecture. The super-net consists of all the possible modules of the architectures in the search space. To train a super-net for weight-sharing search, a substructure is uniformly sampled and separated from the super-net in each training iteration. The weights are inheritance from the super-net, and only the weights of the selected substructure will be trained in this iteration.

Specially, we adapt the algorithm in Alg.1 to find the best architecture in the trained super-net. The only difference here is that we do not need to train and evaluate an architecture from scratch; instead we use the substructure separated from the super-net, run a large validation batch and calculate the accuracy as the evaluation of the architecture.

## 6 EXPERIMENTS

In this section, we evaluate the effectiveness and generality of the GOAL method. The code for our approach is provided for research purpose.[1]

### 6.1 DATASETS

We run experiments on two NAS benchmark datasets: *NAS-Bench-101* and *NAS-Bench-201* (Ying et al., 2019; Dong & Yang, 2020). The datasets obtain validation and testing accuracy of all architectures in their search spaces under consistent training and testing settings. We use the validation and test accuracy of the image classification task on CIFAR-10, provided by both datasets, as our search target. A brief description of the search spaces provided by the datasets is presented in Appendix.A.

### 6.2 STANDALONE SURROGATE MODEL ANALYSIS

We first evaluate the standalone surrogate model. To evaluate the ability of ranking architectures of the proposed surrogate, we train the model to rank the architectures in NAS-Bench-201 by the validation accuracy. We split 3k out of 6k unique structures in NAS-Bench-201 as the test set. For the training set, we sample $\{128, 256, 512\}$ architectures with their final validation accuracy to evaluate the models with different sizes of training data. The GNN model consists of 4 ECC layers of 64 hidden layer size. For comparison, we build an alternative multilayer perceptron model following AlphaX (Wang et al., 2019b), which is stacked by 4 layers of 64 hidden units. We train each model with a batch size of 32 and Adam optimizer with $2 \times 10^{-3}$ learning rate and $5 \times 10^{-4}$ weight decay. We use the Spearman's rank correlation coefficient $\rho$ between the ground truth and the predicted score as the evaluation metric. We rank the results in ascending order of validation error, thus the lower rank order donates a better architecture.

As shown in Fig.3, the GNN model suppresses the MLP model constantly. Note that during the actual searching process, training data of the surrogate is dynamically selected by gradient descent. We present the surrogate performance in such a situation in Appendix.D.

---

[1]Code is included with the supplementary material. Code will be released upon the paper acceptance.

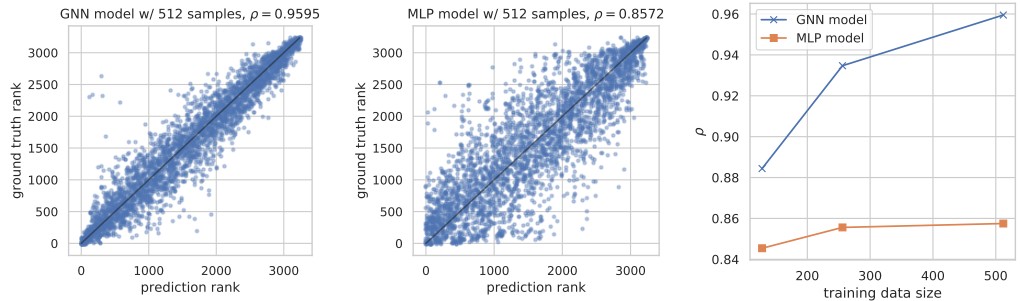

Figure 3: Ranking results on architectures in *NAS-Bench-201*, where $\rho$ stands for Spearman's rank correlation coefficient, lower rank order stands for better architecture. Left: ranking results of different models; right: comparison between models with varying training data sizes.

## 6.3 SEARCH RESULTS

**Settings**  For search evaluation, we follow the original benchmark protocols of *NAS-Bench-101* and *NAS-Bench-201*. We use the final validation accuracy of *NAS-Bench-101* and the 12th training epoch validation accuracy of *NAS-Bench-201* as the search signal, then report the final valid and test accuracy of the searched architectures. Both the validation and test accuracy are provided by the benchmark datasets. We keep the stochastic accuracy setting of both datasets, which returns a random accuracy value for each query to simulate the randomness in architecture training and evaluation. We run all algorithms for 50 trails with different random seeds, then take the mean and standard deviation of the results. We set the terminating condition of all methods as the max budget for the total training time of all evaluated architectures, for $5 \times 10^5$ seconds on *NAS-Bench-101* (for about 300 samples) and $2.8 \times 10^4$ seconds on *NAS-Bench-201* (for about 240 samples). For experiments on GOAL, we set $K = 5$, gradient step size $\mu = 1$, initial pool size $N = 64$ for *NAS-Bench-101*, $N = 32$ for *NAS-Bench-201*. The surrogate model $f$ is trained for 25 epochs for each iteration in Alg.1. Other model settings keep the same as in Sec.6.2.

**Baselines**  We use the following state-of-the-art baselines for comparison on both datasets: ● *Random Search (RS)* (Ying et al., 2019; Dong & Yang, 2020). ● *Reinforcement Learning (RL)* (Zoph & Le, 2016; Ying et al., 2019). ● *Regularized Evolution Algorithm (REA)* (Real et al., 2019). ● *Bayesian Optimization Hyperband (BOHB)* (Falkner et al., 2018). ● *Sequential Model-based Algorithm Configuration (SMAC)*. ● *Neural Predictor for Neural Architecture Search (NPNAS)* (Wen et al., 2019). For fair comparison, we equip NPNAS with the GNN surrogate model we proposed.

To validate the contribution of GNN surrogate in GOAL, we additionally includes a variant, namely GOAL-MLP, which employs MLP model instead of GNN as the surrogate model in the GOAL pipeline.

On *NAS-Bench-101*, we additionally run *AlphaX* (Wang et al., 2019b) and *Neural Architecture Optimization (NAO)* (Luo et al., 2018).

We provide detailed settings of the baselines in Appendix.B.

**Results**  Fig.4 shows the mean accuracies of the searched best architectures of the compared algorithms. We also appear the version with standard deviation error bar in Appendix.C. We show both validation and test curve with the total time cost growing. The *Oracle* on *NAS-Bench-201* shows the upper bound, which indicates the average and variance of the results of the global optimal architectures selected by the best 12th valid accuracy.

GOAL shows promising efficiency on both datasets. Comparing to the small search space of *NAS-Bench-201*, advantages of GOAL are more obvious on *NAS-Bench-101*, showing that our method is more efficient to handle complex graph structures in large search space. In the smaller search space of *NAS-Bench-201*, our GOAL quickly assess the range of *Oracle* after about $1.2 \times 10^4$ seconds, and robustly stays at the global optima after $1.7 \times 10^4$ seconds, showing that GOAL can quickly find the global optimal solutions on small search space.

Note that though GOAL-MLP marginally better than other baselines, the comparison between GOAL and GOAL-MLP shows the importance of the high-quality graph representation of GOAL. However, both GOAL-MLP and GOAL significantly suppress NPNAS, the offline GNN predictor based method, and NAO, the encoder-decoder based optimization method, shows that our proximal optimization based pipeline can improve the exploitation efficiency.

A further case study of the searched architectures is presented in Appendix.E.

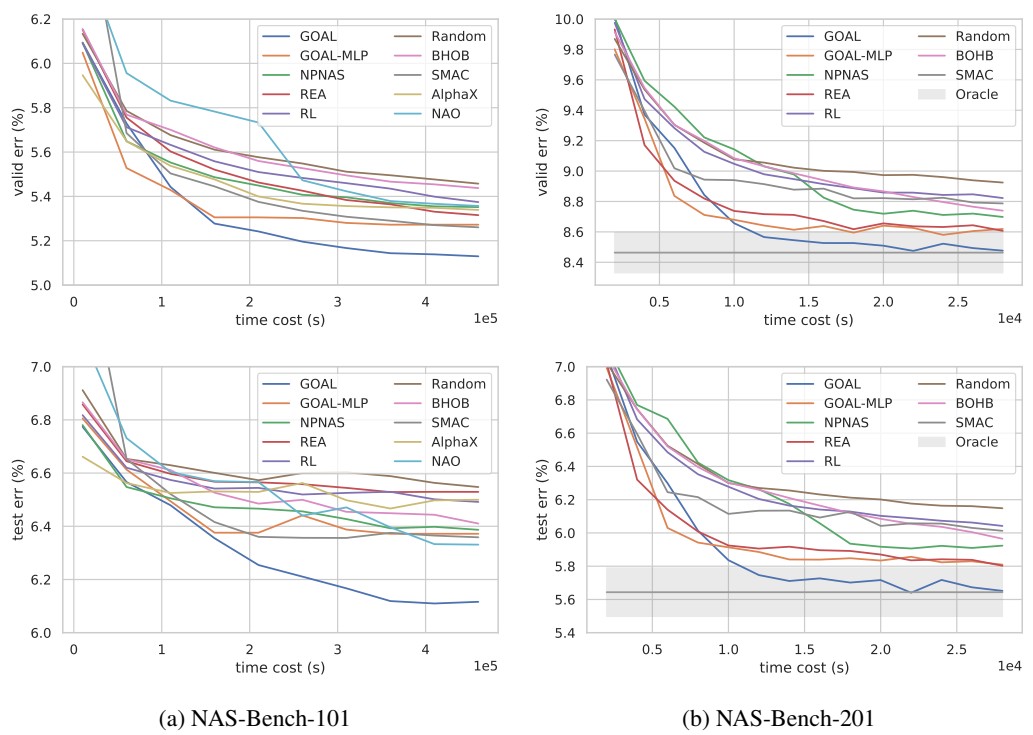

(a) NAS-Bench-101                (b) NAS-Bench-201

Figure 4: Comparisons of search efficiency. Left: validation (top) and test (bottom) error of architectures searched on NAS-Bench-101; right: validation (top) and test (bottom) error of architectures searched on NAS-Bench-201.

### 6.4 WEIGHT-SHARING RESULTS

**NAS Bench 201 Space** We run the weight-sharing search on *NAS-Bench-201* to evaluate the performance of GOAL with weight-sharing strategy. Following the settings in *NAS-Bench-201*, we train a super-net for 250 epochs on CIFAR-10, then run the search process for 4 trails with different seeds to evaluate the mean performance. For comparison, we use random-search (Li & Talwalkar, 2019) and SPOS (Guo et al., 2019), state-of-the-art based on evolution algorithm, as baselines. As shown in table 1, GOAL leads better and more robust search results comparing to the baselines, suggesting the fitness with weight-sharing strategy.

**DARTS Search Space** To validate GOAL on larger search space, we perform weight-sharing search in the DARTS space (Liu et al., 2019) on CIFAR-10 dataset. We use the same size of proxy super-net as DARTS, following the super-net training settings in Li & Talwalkar (2019). For GOAL search stage, we set $K = 16$, initial pool size $N = 256$, gradient step size $\mu = 1$. We totally evaluated 1024 architectures in the search stage. The found architecture is validated on CIFAR-10 following the settings in Liu et al. (2019).

The final performance is summarized in Table 2. GOAL finds more competitive architecture in identical search space comparing to other weight-sharing methods based on different search strategies. The architecture searched by GOAL is visualized in Appendix.F.

Table 1: Results on weight-sharing search on *NAS-Bench-201*. [†] Reported by Dong & Yang (2020). DARTS failed on *NAS-Bench-201* benchmark due to its sensibility on search space and hyperparameters. [‡] 100 samples evaluated on 250 epoch trained super-net.

| Method | CIFAR-10 validation (%) | CIFAR-10 test (%) |
|---|---|---|
| DARTS[†] (Liu et al., 2019) | 39.77±0.00 | 54.30±0.00 |
| Random[†‡] (Li & Talwalkar, 2019) | 84.16±1.69 | 87.66±1.69 |
| SPOS[‡] (Guo et al., 2019) | 88.91±0.97 | 92.32±0.89 |
| GOAL[‡] | 89.33±0.64 | 92.84±0.51 |

Table 2: DARTS space result on CIFAR-10 dataset.

| Architecture | Test Err. (%) | Search Method |
|---|---|---|
| ENAS (Pham et al., 2018) | 2.89 | RL |
| NAO-WS (Luo et al., 2018) | 2.93 | NAO |
| DARTS (Liu et al., 2019) | 2.76 | gradient |
| Random (Li & Talwalkar, 2019) | 2.85 | random |
| GOAL | 2.64 | GOAL |

## 7 CONCLUSION

In this paper, we proposed Graph Optimized Neural Architecture Learning (GOAL), a novel gradient-based approach for neural architecture search combined with graph structure learning. Different from popular black-box optimization based methods, we explicitly model the NAS as the optimization on graph structure, and apply proximal gradient descent on discrete graph structures with assistance of a GNN-based surrogate model. Our GOAL outperforms mainstream SOTA NAS and hyper-parameter optimization methods on NAS benchmarks, shows promising efficiency on the NAS problem.

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

# A    SEARCH SPACES OF THE DATASETS

The *NAS-Bench-101* and *NAS-Bench-201* form their search space differently:

- *NAS-Bench-101* donates graph nodes as neural operations, and graph edges as input/output connection, as the left of Fig.2 shows. The dataset provides a search space with at most 7 nodes and 9 edges. Except for *input* and *output* nodes, intermediate nodes are one of the operations in $\{1 \times 1$ convolution, $3 \times 3$ convolution, $3 \times 3$ max pooling$\}$. There are totally 423,624 unique graphs in the search space.

- *NAS-Bench-201* uses graph edges to represent both connections and operations, nodes to aggregate intermediate outputs, just like the right of Fig.2. The nodes donate tensor element-wise addition, while edges are operations in $\{1 \times 1$ convolution, $3 \times 3$ convolution, $3 \times 3$ average pooling, skip connection, zeroize$\}$. This search space consists of 15,625 unique models without considering graph isomorphism. After de-duplicating by graph isomorphism, there are 6,466 unique topology structures.

# B    DETAILED BASELINE SETTINGS

Following SOTA of NAS and hyper-parameter optimization methods are used in both *NAS-Bench-101* and *NAS-Bench-201* experiments:

- **Random Search (RS)** (Ying et al., 2019; Dong & Yang, 2020). As reported by many prior works, RS is a competitive algorithm for NAS. We use the implements and settings in Ying et al. (2019) and Dong & Yang (2020).

- **Reinforcement Learning (RL)** (Zoph & Le, 2016; Ying et al., 2019). For the RL method, we use the implementation in (Ying et al., 2019), which directly learns the architecture distribution as action policy, instead of learning an RNN controller as in (Zoph et al., 2017).

- **Regularized Evolution Algorithm (REA)** (Real et al., 2019). The original benchmarks of both datasets show that REA is the state-of-the-art algorithm among the benchmarks (Ying et al., 2019; Dong & Yang, 2020). We inherit the settings from Ying et al. (2019); Dong & Yang (2020).

- **Bayesian Optimization Hyperband (BOHB)** (Falkner et al., 2018). BOHB is a Bayesian optimization method combined with hyperband for fast hyperparameter optimization. We inherit the settings from Ying et al. (2019); Dong & Yang (2020).

- **Sequential Model-based Algorithm Configuration (SMAC)** (Hutter et al., 2011). SMAC is a sequential model-based Bayesian optimization method which employs random forest as the predictor. We inherit the settings from Ying et al. (2019).

- **Neural Predictor for Neural Architecture Search (NPNAS)** (Wen et al., 2019). The NPNAS samples fixed size of architectures from the search space as training data for training a GCN-based surrogate model for once, then freeze the surrogate model and predict the performance of all architectures in the search space to select the best ones. We use the surrogate model we proposed to imitate the method of NPNAS for fair comparison on the searching phrase. We set the training set size to 172 on *NAS-Bench-101* and 128 on *NAS-Bench-201*, which are reported as the optimal values for our search constraint by (Wen et al., 2019).

Following baselines are additionally run on *NAS-Bench-101*:

- **Neural Architecture Optimization** (Luo et al., 2018). NAO employs an auto-encoder and perform searching in the learnt embedding space of architectures. We follow the NAS-Bench settings in Luo et al. (2020).

- **AlphaX** (Wang et al., 2019b). AlphaX adapts monte-carlo tree search on NAS problem. We keep the settings unchanged from Wang et al. (2019b).

# C    EXPERIMENT RESULTS WITH VARIANCE

The experiment results on *NAS-Bench-101* and *NAS-Bench-201* with standard deviation is shown in Fig.5. Note that the variance of the test error on *NAS-Bench-101* is higher than the variance of validation error, since our direct search target is the validation error, and the correlation between validation error and test error is not perfect. Comparing to other baselines, the better mean results of GOAL do not cause higher variance, or even lower, showing the efficiency of GOAL is robust and promising.

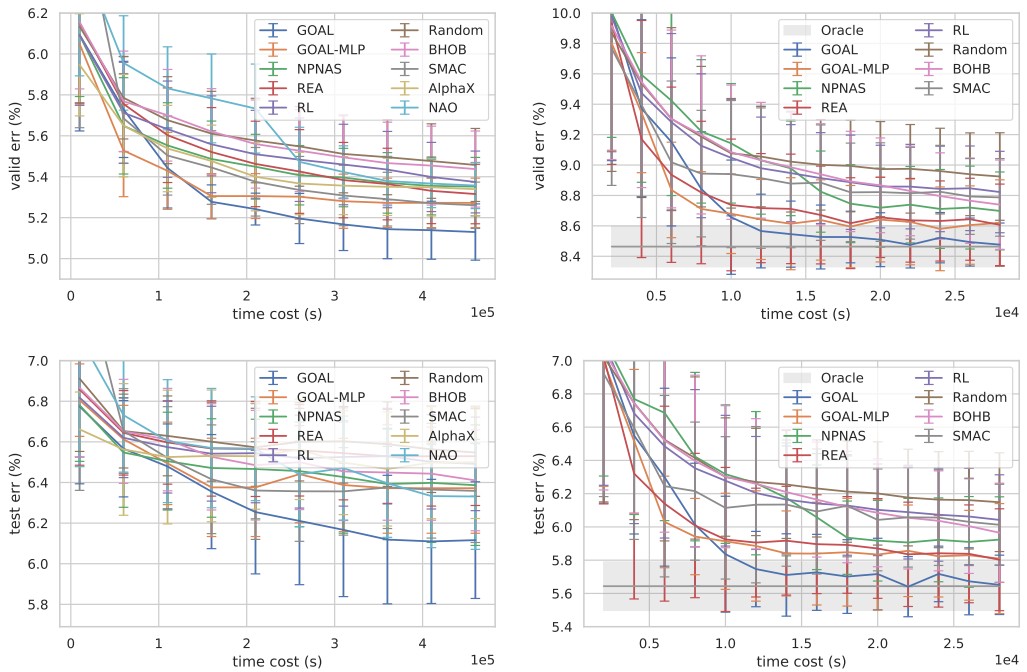

Figure 5: Comparisons of search efficiency with standard deviation. Left: validation (top) and test (bottom) error of architectures searched on NAS-Bench-101; right: validation (top) and test (bottom) error of architectures searched on NAS-Bench-201.

# D    SURROGATE MODEL PERFORMANCE ACCOMPLISHED SEARCHING

To analysis the behavior of our surrogate model in the search process, we show the ranking results on all unique architectures in *NAS-Bench-201* of the surrogate model accomplished search process in Fig.6a. For comparison, we also retrained the model with uniformly sampled training data and show the results in Fig.6b. Comparing to the ranking results in Fig.6b, the model accomplished searching performs better on the left-bottom corner, which consists of the best models in the whole search space. Since the architectures with good performance are what we actually concern in the NAS problem, our alternately training strategy in the searching process could enhance the model comparing with the training-once approaches like NPNAS.

# E    CASE STUDY OF THE ARCHITECTURE TRANSITION

Fig.7 shows the evolution history of an architecture found by GOAL on *NAS-Bench-101*. The case starts at an extremely bad initialization, which directly sends the input to the output. The first step adds several convolution nodes in the company of ResNets-like skip-connections, leading to a meaningful structure. The following steps shrink the architecture to be smaller and fine-tune the connections. As presented in the transition steps, in the *NAS-Bench-101* space, more operations do not always lead to better performance, and the connectivity impacts lot on the final validation

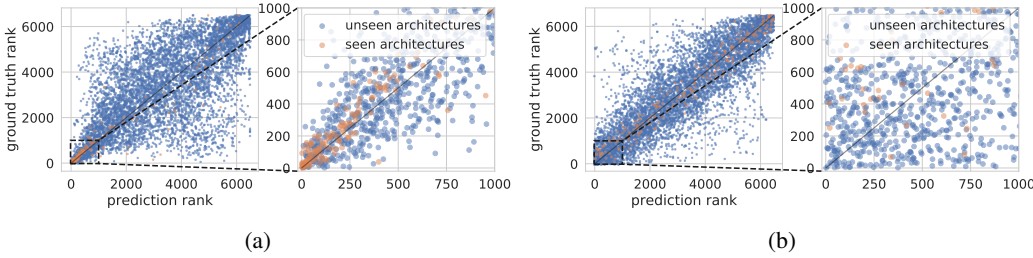

(a)               (b)

Figure 6: Left: the ranking result of the model accomplished search process on *NAS-Bench-201*; right: the ranking results of the model retrained with the same size of uniformly sampled training data.

accuracy. Notice that only $3 \times 3$ convolution operations are selected in these architectures, which shows that other operations like max pooling are less helpful for improving accuracy on the tasks in *NAS-Bench-101*.

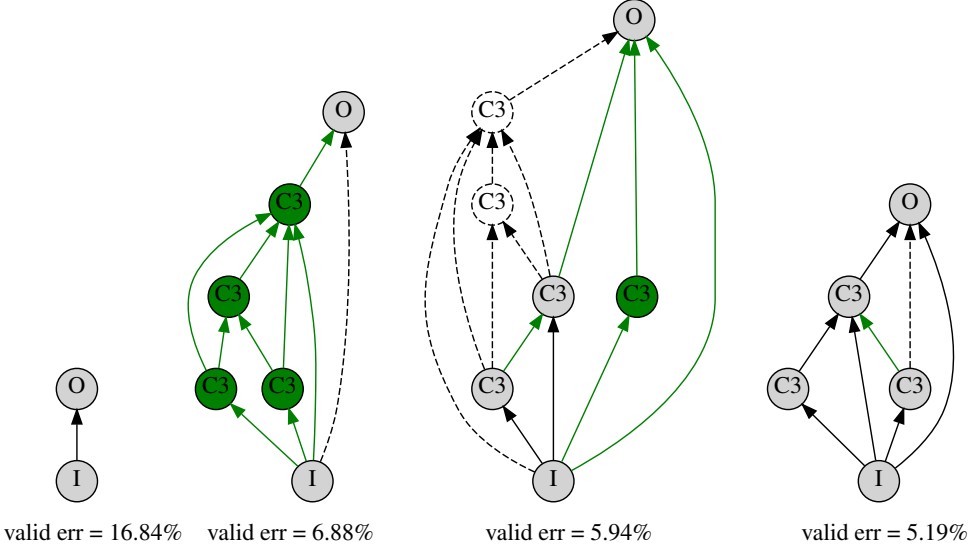

valid err = 16.84%  valid err = 6.88%    valid err = 5.94%    valid err = 5.19%

Figure 7: The 3-step evolution history of a found architecture on *NAS-Bench-101*. 'I', 'O', and 'C3' mean for input node, output node and $3 \times 3$ convolution node respectively. The green parts are newly added, while dashed parts indicate the deletion.

# F  ARCHITECTURE SEARCHED ON DARTS SPACE

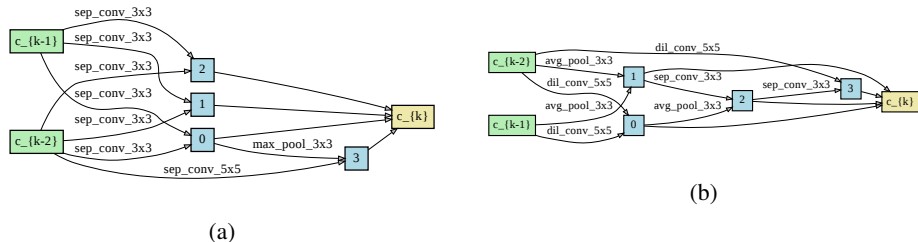

(a)               (b)

Figure 8: Normal (left) and reduction (right) cell found by GOAL on DARTS space.

