# OpenReview forum: "Differentiable Graph Optimization for Neural Architecture Search"
_ICLR.cc/2021/Conference — Reject_

### Official Review · AnonReviewer3 · 2020-10-15
**Novel but should compare with NAO and evaluate on ImageNet or CIFAR10.**

**Rating:** 5
**Confidence:** 4

**Review:**

#### Summary:
This work propose Graph Optimized Neural Architecture Learning, that uses a differentiable surrogate model to directly optimize the graph structures. More specifically, the surrogate model takes a graph structure as the neural architecture embedding and predicts a relative ranking, then applies gradient descent on the input graph structure to optimize the neural architecture. GOAL demonstrates superior performance compared to SoTAs.

#### Weakness:

-First, the method is quite similar to NAO, where an encoder and decoder approach the maps neural architectures into a continuous space and builds a predictor based on the latent representation. The only difference is that NAO is use a decoder to decode the optimized latent representation back to architecture representation while here GOAL applies gradient descent on a graph neural network. Also, how to decode the parameters back to the neural architecture discrete representation is not clearly explained in the paper.

-Second,  this method can work well for small models and small search spaces, but can be hardly applied to larger models. Training the surrogate function for a larger search space or larger models can take more samples and more training time (e.g. large models takes much longer time to train, thus even a proxy accuracy should take more time to evaluate). A parameter sharing scheme can be very inaccurate in the beginning therefore results in suboptimal architecture selection. The inaccuracy is compounded when using the same model for both a surrogate function and neural architecture search. The evaluation on only NAS-bench partially verifies the reviewers concerns.

#### Detailed feedback:
The reviewer would like to suggest several fixes to this paper:
-First, try to compare to better baselines (e.g. NAO, more recent differentiable search work) on not only NASBench, but on real CIFAR10 or ImageNet workloads.

-Second, compare a non graph neural network based approach with GOAL and show the necessity of using a graph neural network.

-Third, be more clear about decoding back the graph neural network parameterizations to the neural architecture representation. Include more details on the number of samples used to train the surrogate function and hyperparameteers used in algorithm 1.


[1] "Neural Architecture Optimization", https://arxiv.org/abs/1808.07233

---

> ### Author Response · Authors · 2020-11-18
> **Response to Reviewer 3**
>
> We would like to thank the reviewer for your careful reading and providing a lot of valuable comments! Below we address the concerns mentioned in the review:
>
> Response to the concerns:
>
> Q1. Differences and comparison to NAO
>
> NAO [1] aims to map neural architecture to latent space to perform continuous optimization. This requires a learnt encoder and decoder to convert between latent representation and architecture. Our motivation is developed from another direction, in contrast, which aims to optimize discrete graph structure.
>
> Since we didn’t introduce latent space like NAO in the optimization steps, GOAL doesn’t require a particular decoder model besides the quantization function $q$ described in Sec.5.1. What the quantization function $q$ behaves is to round the relaxed categorical parameters to the nearest one-hot categories, which means just take the largest edge and node selections here. For NAS-Bench-101, we set K= 5, μ= 1, N= 64; for NAS-Bench-201, we set K= 5, μ= 1, N= 32.
>
> Unlike a learnt decoder, the quantization function is quite guaranteed since the output of a learnt decoder is unexpectable and depends on the quality of training. The empirical result in Fig.4 shows that GOAL constantly suppresses NAO on NAS-Bench-101. We also added more comparison between NAO on larger search spaces, please refer to (2).
>
> Q2. Larger search spaces
>
> To check the ability of GOAL for larger search spaces, we added a new experiment on DARTS search space in the parameter sharing scheme. For supernet training, we use the same size of proxy model from DARTS, and follow the architecture sample method from [2]. We keep other hyper-parameters same as DARTS in training, only increasing the number of training epochs to 150 following [2]. In the search episode we totally evaluate 1024 architectures using GOAL pipeline, while K=16 and N=256 is setted to fit the larger search space.
>
> Method   |   CIFAR10 error
>
> NAO-WS [1] | 2.93
>
> DARTS [3] | 2.76
>
> GOAL-WS | 2.64
>
> The result shows that GOAL can work well for larger search space like DARTS under parameter sharing scheme, since the architecture found by GOAL is better than baselines of NAO and DARTS. We admit that a parameter sharing scheme can result in inaccurate proxy accuracy, however, as shown in prior works [2,3], improving the search method can still be positive for parameter sharing search.
>
>
> Response to the suggestions:
>
> - Compare to better baselines
>
>   Please refer to the concern 2.
>
> - Compare a non graph neural network based approach
>
>   Please refer to the response to R2.(4).
>
> - Be more clear about decoding back the graph neural network parameterizations
>
>   Please refer to the concern 1.
>
>
> [1] Neural Architecture Optimization
>
> [2] Random search and reproducibility for neural architecture search
>
> [3] Single Path One-Shot Neural Architecture Search with Uniform Sampling

---

### Official Review · AnonReviewer4 · 2020-10-27
**Simple method with promising performance; experiments not fully convincing**

**Rating:** 6
**Confidence:** 3

**Review:**

The authors address the Neural Architecture Search problem. At the core of their contribution is an architectural improvement; performance prediction of a considered architecture is much better when using a particular graph neural network on (softened) architecture topology. The rest of the NAS pipeline is naturally built around this observation and the final performance looks quite strong.

I think the topic is of sufficient significance to the community and I find the proposed method well-tailored to the problem and sufficiently original. Also, the writeup is easy to follow (up to minor issues listed below).

Since the contribution is mainly in finding a good architecture for the surrogate model, I find it very unsatisfying that all experiments are carried out on CIFAR-10. The ML community has had some bad experience with architectures overfitting to datasets and more thorough evaluation is needed. Also, the prime purpose of the NAS line of work isn't to find a good architecture for CIFAR-10, and if this method aspires to have a broader impact in the NAS community (which I think it should), the experimental section needs to be expanded.

Minor points:
Eq 3: I don't understand. Due to the $\textrm{sign}$ function, the loss does not seem differentiable w.r.t $\tilde{y}_i$. Or do the brackets need some regrouping?
The end of Sec 5.1: Currently $\tau$ cancels in the definition of $q$. I guess it should be inside the brackets. Also, what does it mean to "practically" take a limit? The point seems to be that softmax with very low temperature reduces to a hard one-hot vector but I do not understand how the authors use this precisely. Why not "round" to the one-hot vector right away?
Sec 2.2: models -> model
Sec 4.1 donated -> denoted?
Experiments: One ablation that I think would be great for giving insight to inner working of the method would be the following: In Figure 4, include also the performance curve of GOAL with the MLP architecture for rank prediction. This should nicely demonstrate where your main technical point is.

---

> ### Author Response · Authors · 2020-11-18
> **Response to Reviewer 4**
>
> We are grateful to the reviewer for a nice summary, and for the kind recognition of our key contributions. Below we address the concerns mentioned in the review:
>
> Q1. “Overfitting on CIFAR-10”
>
> We agree that NAS is not aiming to overfit one dataset. We tried to avoid such fashion by applying the two different widely used NAS-Bench benchmarks, which provides their own search space and corresponding accuracies, instead of directly touching the CIFAR-10 dataset. This could ensure a fair comparison and somehow prevent from falling into overfitting. Evaluation on widely usage and more tasks requires non-trivial adjustments like search space designing, we will leave it for the feature work.
>
> Q2. Minor points.
>
> Q2a: Eq 3: I don't understand.
>
> Sorry for the confusion. The sign function is applied on (y_i - y_j), which is the ground-truth value and determines the direction of optimization for this pair. The output value \tilde{y} is not applied by sign function. We rewrite this formulation in the revised manuscript for clearer expression.
>
> Q2b: The end of Sec 5.1
>
> Yes, we do mean to “round” the vector to the one-hot vector. Thanks for the correction and we improved the statements in the revised manuscript.
>
> Q2c: Typos
>
> We have fixed them in the revision.
>
> Q2d: Ablation experiments
>
> Thank you for your suggestion, we added a new baseline GOAL-MLP as an ablation of the GNN part in the revised Fig.4 in the updated submission, which uses the optimization steps of GOAL but employs MLP as the surrogate model instead of GNN. The result shows that our pipeline without GNN can be slightly better than other baselines, but be not as good as GOAL with reliable graph representation.

---

### Official Review · AnonReviewer1 · 2020-11-03
**Reasonable approach with limited novelty**

**Rating:** 4
**Confidence:** 4

**Review:**

The paper proposed (1) to use a graph neural net to predict the performance of network architectures, (2) to query new architecture proposals to try next from the predictor, and (3) to iteratively refine the predictor using the collected dataset using gradient descent. Overall, the method is well-motivated and the paper is easy to follow.

My main concerns are as follows:
* In section 2.1 the authors claim that "In contrast, our method directly optimizes the discrete architectures, avoids the drawbacks of continuous relaxation". This is not true—the method still relies on continuous relaxation because of \bar{\alpha} in equation (4), and is still subject to discretization discrepancies due to the quantization function q.
* Predictive methods with GCNs are not new, especially considering that Wen et al., 2019 also trained GCN using gradient descent (although in an offline fashion). The only additional ingredients here seem to be (a) interleaving GCN with the data collection process in an online manner, and (b) weight sharing. However, the advantage of (a) has not been empirically verified with ablation studies and (b) is already common nowadays.
* It seems a bit unconventional that the authors reported the best architecture in the main paper but the mean & variance in the appendix. IMO Figure 5 in Appendix C provides a much more accurate picture of the usefulness of different algorithms than Figure 4. In practice, it can be tricky to tell the "best" architecture for new tasks where test sets are not available.
* The improvements are within the range of variance as compared to SPOS. While the results are still overall positive, it may not well justify the additional implementation complexity of the method.

---

> ### Author Response · Authors · 2020-11-18
> **Response to Reviewer 1**
>
> We want to thank the reviewer for your careful reading and providing a lot of critical comments! Below we address the concerns mentioned in the review:
>
> Q1. The claim on discrete is not true
>
> Thanks for pointing out our inaccurate expression. The “discrete” here means in our pipeline we applied proximal optimization to find the solution on the discrete feasible space, which is different from other approximal methods which optimize in a continuous space [1][2]. The continuous relaxation is introduced in the proximal update steps, the architecture still keeps discrete for the surrogate model, avoiding the discretization discrepancies in architecture criterion. We have modified the description here in the revised version to clarify the confusion.
>
> Q2. “Predictive methods with GCNs are not new ...”
>
> We admit that predictive GCN models are already introduced by previous work like [3], which we have mentioned in the paper. To highlight the differences between our method and [3], there are two points worth noting here.
>
> First of all, as the reviewer mentioned, the motivation behind our GCN model is not simply making predictions, but providing new architecture proposals to try next. The “data collection process in an online manner” is non-trivial and important, since it’s impossible to enumerate all possible proposals as [3] when the search space is large combinatorial graph structures.
>
> Second, the online manner can continuously improve the quality of the predictor during the search procedure, resulting in better exploitation in the search. In Figure 4, we compared GOAL to NPNAS, which is the offline method from [3] equipped with the predictor we proposed. The empirical result shows that our pipeline outperforms their method in the offline manner, demonstrating the effectiveness of our design. In Appx.D Fig.6, we showed the different prediction quality of online-trained model and offline-trained model, which tells that online-trained model makes more accurate predictions on the good candidates.
>
>
> Q3. “It seems a bit unconventional that the authors reported the best architecture in the main paper but the mean & variance in the appendix.”... In practice, it can be tricky to tell the "best" architecture for new tasks where test sets are not available.
>
> Sorry for the confusion. Figure 4 is identical with Appx.C Figure 5, only omitting the variance bar for clearer representation on the trends. We agree that the variance information is important, so we add the version with the variation bar in the appendix for anyone who’s interested.
>
> The “best” architecture in the search procedure is selected by the validation (or dev) set accuracy, as the direct search signal we described in Algo.1, not the test set result.
>
> Q4. The improvements are within the range of variance as compared to SPOS. While the results are still overall positive, it may not well justify the additional implementation complexity of the method.
>
> SPOS is a weight-sharing method based on the evolutionary algorithm. We compared GOAL to evolutionary method (REA) without weight-sharing in Figure 4, which already showed that GOAL significantly suppresses evolutionary method. We additionally compared GOAL to SPOS to verify GOAL can still benefit the weight sharing search. The result of GOAL is better than SPOS both on mean and variance.
>
> On complexity, our method is easy to implement using existing GNN toolkits and automatic differentiation tools. We also provided implementation code in supplementary material. As for computation complexity, since the graph size is quite small in the NAS search spaces (under 10 nodes), the overhead of GNN computing is subtle compared to the training and evaluation cost in NAS.
>
> [1] Neural Architecture Optimization
>
> [2] DARTS: Differentiable Architecture Search
>
> [3] Neural predictor for neural architecture search

---

### Decision · Program_Chairs · 2021-01-07
**Final Decision**

**Decision:**

Reject

**Comment:**

This work was deemed interesting by the reviewers, but they highlighted the following weaknesses in this version of the paper:

- Lack of comparison to other methods.

- Lack of novelty compared to previous work.

- Fundamental problem with training only on one dataset (MNIST), issue with possible overfitting.